# iSplit LBI: Individualized Partial Ranking with Ties via Split LBI

**Qianqian Xu**[1]     **Xinwei Sun**[2]     **Zhiyong Yang**[3,4]
**Xiaochun Cao**[3,4,7]   **Qingming Huang**[1,5,6,7]   **Yuan Yao**[8]

[1]Key Lab. of Intelligent Information Processing, Institute of Computing Technology, CAS
[2]Microsoft Research Asia
[3]State Key Laboratory of Information Security, Institute of Information Engineering, CAS
[4]School of Cyber Security, University of Chinese Academy of Sciences
[5]School of Computer Science and Tech., University of Chinese Academy of Sciences
[6]Key Laboratory of Big Data Mining and Knowledge Management, CAS
[7]Peng Cheng Laboratory
[8]Department of Mathematics, Hong Kong University of Science and Technology
xuqianqian@ict.ac.cn, xinsun@microsoft.com, yangzhiyong@iie.ac.cn
caoxiaochun@iie.ac.cn, qmhuang@ucas.ac.cn, yuany@ust.hk

## Abstract

Due to the inherent uncertainty of data, the problem of predicting partial ranking from pairwise comparison data with ties has attracted increasing interest in recent years. However, in real-world scenarios, different individuals often hold distinct preferences. It might be misleading to merely look at a global partial ranking while ignoring personal diversity. In this paper, instead of learning a global ranking which is agreed with the consensus, we pursue the tie-aware partial ranking from an individualized perspective. Particularly, we formulate a unified framework which not only can be used for individualized partial ranking prediction, but also be helpful for abnormal user selection. This is realized by a variable splitting-based algorithm called `iSplitLBI`. Specifically, our algorithm generates a sequence of estimations with a regularization path, where both the hyperparameters and model parameters are updated. At each step of the path, the parameters can be decomposed into three orthogonal parts, namely, abnormal signals, personalized signals and random noise. The abnormal signals can serve the purpose of abnormal user selection, while the abnormal signals and personalized signals together are mainly responsible for individual partial ranking prediction. Extensive experiments on simulated and real-world datasets demonstrate that our new approach significantly outperforms state-of-the-art alternatives.

## 1   Introduction

The flourish of various online crowdsourcing services (e.g., Amazon Mechanical Turk), presents us an effective way to distribute tasks to human workers around the world, on-demand and at scale. Recently, there arises a plethora of pairwise comparison data in crowdsourcing experiments on the Internet [16, 2], ranging from marketing and advertisements to competitions and election. Information of this kind is all around us: which college a student selected, who won the chess match, which movie a user watched, etc. How to aggregate the massive amount of personalized pairwise comparison data to reveal the global preference function has been one important topic in the last decades [4, 11, 26, 20, 2, 22].

*But is the aggregated result necessarily more important than individual opinions?* This is not always the case especially when our Internet is flooded with personalized information in diversity. The disagreement over the crowd could not be simply interpreted as a random perturbation of a consensus that everybody should follow. For example, we often observe quite different preferences on a college ranking or a favorite movie list. Hence the wave of personalized ranking arises in recent years in search of better individualized models. One line of the related research assumes that the ranking function is determined by a small number of underlying intrinsic functions such that every individual's personalized preference is a linear combination of these intrinsic functions [30, 19, 12, 6]. Another line of research attributes the personalized bias to user quality, where either a single parameter or a general confusion matrix is adopted to model the users' ability to provide a correct label [7, 13, 15, 23, 25, 33]. There is also a trend to explore personalized ranking effects in terms of preference distributions [18, 17]. Moreover, [28, 27] take a wide spectrum by considering both the social preference and individual variations simultaneously. Specifically, it designs a basic linear mixed-effect model which not only can derive the common preference on population-level, but also can estimate user's preference/utility deviation in an individual-level.

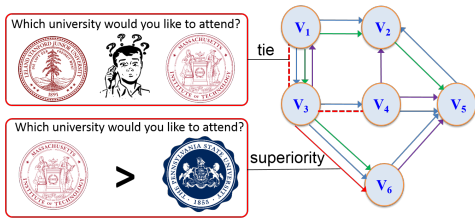

Figure 1: An example of pairwise ranking with ties.

All the work mentioned above either focuses on instance-wise preference learning or assumes that the candidates are comparable in a total order. For pairwise preference learning, however, the answer might go beyond a win/loss option in real-world scenarios. The following gives an example in crowd-sourced college ranking.

**Example**. *In world college ranking with crowdsourcing platforms such as* Allourideas*, a participant is asked about "which university (of the following two) would you rather attend?". As is shown in Fig.1, let $G = (V, E)$ be a pairwise ranking graph whose vertex set is $V$, the set of universities to be ranked, and the edge set is $E$, the set of university pairs which receive some comparisons from users. Here different colors indicate different users. If a voter thinks college $V_3$ is better than college $V_6$, a solid arrowed line from $V_3$ to $V_6$ occurs (i.e., superiority). However, when a voter thinks the two colleges (i.e., $V_1$ and $V_3$) listed are incomparable and difficult to judge, he may click the button "I can't decide", then a dotted line connecting $V_1$ and $V_3$ happens (i.e., tie).*

Here for a pair $(i, j)$, if a voter believes $i$ and $j$ share a similar strength and neither one is superior to the other, he may abstain from this decision and leave it with a tie. An abstention of this kind is an obvious means to avoid unreliable predictions. Such kind of pairwise comparison data, together with "I cannot decide" decision, provide us information about possible ties or equivalent classes of items in partial orders. Though there is some work in the literature studying how to organize information in partial orders of such tied subsets or equivalent classes (partitions, bucket orders) [5, 14], little has been done on learning the *individualized* partial order models from such pairwise comparison data with ties.

In this paper, we aim to learn the individualized partial ranking models for each user based on such kind of pairwise ranking graph with ties. Based on the partial ranking, we could recommend universities for a specific user. For example, recommending universities that are with the same quality as college A; or, recommending universities that are slightly better than college B, etc.

Moreover, another challenge of personalized preference ranking comes from the fact that abnormal users might exist in the crowd. They either bear an extremely different pattern with the majority of the crowd or belong to malicious users trying to attack the learning system. To deal with abnormal user detection in crowdsourced data, existing studies often take a majority voting strategy, which often ignores the personalized effect.

Seeing the issues mentioned above, we propose a unified framework, called `iSplitLBI`, for personalized partial ranking, tie state recognition, and abnormal user detection. The merits of our framework are of three-fold: 1) It decomposes the parameters into three orthogonal parts, namely, abnormal signals, personalized signals, and random noise. The abnormal signals can serve the purpose of abnormal user detection, while the abnormal signals and personalized signals together are mainly responsible for user partial ranking prediction. 2) It provides a compatible framework be-

tween predict individual preferences (i.e., *model prediction*) and identification of abnormal users (i.e., *model selection*) by virtue of variable splitting scheme. 3) Exploiting the regularization path, it simultaneously searches hyper-parameters and model parameters. Up to our knowledge, this is the first proposal of such a model in the literature on partial ranking.

## 2  Methodology

In crowdsourced pairwise comparison experiments, suppose there are $n$ alternatives or items to be ranked. Traditionally, the pairwise comparison labels collected from users can be naturally represented as a directed comparison graph $G = (V, E)$. Let $V = \{1, 2, \ldots, n\}$ be the vertex set of $n$ items and $E = \{(u, i, j) : i, j \in V, u \in U\}$ be the set of edges, where $U$ is the set of all users who compared items. User $u$ provides his/her preference between choice $i$ and $j$, such that $y_{ij}^u = 1$ means $u$ prefers $i$ to $j$ and $y_{ij}^u = -1$ otherwise.

However, in real-world applications, ties are ubiquitous. In this case, if a rater thinks neither of the two items in a pair is superior to the other, he/she may abstain from this decision and instead declare a tie, as is shown with the red dotted line in Fig.1. This inspires us to adopt a win/tie/lose user feedback in the following sense:

$$y_{ij}^u = \begin{cases} 1, & \text{if } u \text{ prefers } i \text{ to } j, \\ -1, & \text{if } u \text{ prefers } j \text{ to } i, \\ 0, & \text{otherwise.} \end{cases} \tag{1}$$

Given the definition of the user feedback, in the rest of this section, we elaborate our proposed model in the following order. First we propose a probability model to describe the generation process of the comparison results $y_{ij}^u$. Then we present a simple iterative algorithm called individualized Split Linearized Bregman Iterations (i.e., `iSplitLBI`) for individualized partial ranking. In the end, we provide a decomposition property of `iSplitLBI` which dives deeper into the insights of our proposed model.

### 2.1  Probabilistic Model of Partial Ranking with Ties

Now we describe our dataset at hand with the following notations. Suppose that we have $U$ users and for a specific user $u$, he/she annotates $n_u$ pairwise comparisons. For a specific comparison $(i, j)$, the user provides a label $y_{ij}^u$ correspondingly following (1). We denote the set of all pairwise comparisons available for user $u$ as $\mathcal{O}^u$, and define the label set $\mathcal{Y}^u$ as:

$$\mathcal{Y}^u = \left\{ y_{ij}^u : (i, j) \in \mathcal{O}^u \right\} \tag{2}$$

Then our dataset could be expressed as $\{\mathcal{O}^u, \mathcal{Y}^u\}_{u=1}^U$. We assume that each user has a personalized score list for all items. We denote such true personalized score lists as $\boldsymbol{s}^u = [s_1^u, \cdots, s_{n_{u_i}}^u], \forall u$, where $n_{u_i}$ is the number of items that are available for $u$. Furthermore, for any specific $u$, $\lambda^u$ is a personalized threshold value to be learned for decision. Then, for a specific user $u$, and a specific observation $(i, j)$, we assume that $y_{ij}^u$ is produced by comparing the score difference $s_i^u - s_j^u$ with the threshold $\lambda^u$. Meanwhile, to model the randomness of the sampling and the decision making process, we model the uncertainty of $s_i^u - s_j^u$ with an associated random noise $\epsilon_{ij}^u$ which has a c.d.f $\Phi(t)$. Then, in our model, user $u$ would choose $y_{ij}^u = 1$, if the observed personalized score difference $s_i^u - s_j^u + \epsilon_{ij}^u$ is greater than the threshold $\lambda^u$. To the opposite, if $s_i^u - s_j^u + \epsilon_{ij}^u$ is smaller than $-\lambda^u$, then user $u$ would choose $y_{ij}^u = -1$. Otherwise, $s_i^u - s_j^u + \epsilon_{ij}^u$ has a smaller magnitude than $\lambda^u$, in which case the user would claim a tie. Above all, $y_{ij}^u$ is obtained from the following rule:

$$y_{ij}^u = \begin{cases} 1, & s_i^u - s_j^u + \epsilon_{ij}^u > \lambda^u; \\ -1, & s_i^u - s_j^u + \epsilon_{ij}^u \leq -\lambda^u; \\ 0, & \text{else.} \end{cases} \tag{3}$$

Furthermore, we define two variables $\zeta_k^{u+}$ and $\zeta_k^{u-}$ as :

$$\begin{aligned} \zeta_{ij}^{u+} &= \lambda^u - s_i^u + s_j^u \\ \zeta_{ij}^{u-} &= -\lambda^u - s_i^u + s_j^u \end{aligned} \tag{4}$$

Since $\epsilon_{ij}^u$ is a random variable with a c.d.f $\Phi$, we could then derive the probability to observe $y_{ij}^u = 1, 0, -1$, respectively. Specifically, together with (3) and (4) we have:

$$P\{y_{ij}^u = \ \ \ 1\} = P\{\epsilon_{ij}^u > \zeta_{ij}^{u+}\} = 1 - \Phi(\zeta_{ij}^{u+})$$
$$P\{y_{ij}^u = \ \ \ 0\} = P\{\zeta_{ij}^{u-} \le \epsilon_{ij}^u < \zeta_{ij}^{u+}\} = \Phi(\zeta_{ij}^{u+}) - \Phi(\zeta_{ij}^{u-})$$
$$P\{y_{ij}^u = -1\} = P\{\epsilon_{ij}^u \le \zeta_{ij}^{u-}\} = \Phi(\zeta_{ij}^{u-}).$$

Note that different $\Phi$ could lead to different models. In this paper, we simply consider the most widely adopted Bradley-Terry model: $\Phi(t) = \frac{e^t}{1+e^t}$, while leaving other models for future studies.

## 2.2   Individualized Split LBI

In our framework, we assume the majority of participants share a common preference interest and behave rationally, while deviations from that exist but are sparse. To be specific, we consider the following linear model for annotator's individualized partial ranking:

$$\boldsymbol{s}^u = \boldsymbol{c}_s + \boldsymbol{p}_s^u, \quad \lambda^u = c_\lambda + p_\lambda^u, \quad s_i^u - s_j^u + \epsilon_{ij}^u = (c_{s_i} + p_{s_i}^u) - (c_{s_j} + p_{s_j}^u) + \epsilon_{ij}^u. \tag{5}$$

where (1) $\boldsymbol{c}_s$ and $c_\lambda$ represent the consensus level pattern, in which $\boldsymbol{c}_s$ is the common global ranking score, $c_\lambda$ is the common $\lambda$, as a fixed effect, $c_{s_i}$ and $c_{s_j}$ are the $i$th and $j$th element of $\boldsymbol{c}_s$, respectively; (2) $\boldsymbol{p}_s^u$ and $p_\lambda^u$ represent the individualized bias pattern, in which $\boldsymbol{p}_s^u$ is the annotator's preference deviation from the common ranking score $\boldsymbol{c}_s$, $p_\lambda^u$ is the individualized bias with $c_\lambda$, as a random effect, $p_{s_i}^u$ and $p_{s_j}^u$ are the $i$th and $j$th element of $\boldsymbol{p}_s^u$, respectively; (3) $\epsilon_{ij}^u$ is the random noise.

To make the notation clear, let $P_{1,ij}^u = 1 - \Phi(\zeta_{ij}^{u+})$, $P_{0,ij}^u = \Phi(\zeta_{ij}^{u+}) - \Phi(\zeta_{ij}^{u-})$ and $P_{-1,ij}^u = \Phi(\zeta_{ij}^{u-})$, then we could represent $P\{y_{ij}^u\}$ as:

$$P\{y_{ij}^u\} = \left[P_{1,ij}^u\right]^{1\{y_{ij}^u=1\}} \left[P_{0,ij}^u\right]^{1\{y_{ij}^u=0\}} \left[P_{-1,ij}^u\right]^{1\{y_{ij}^u=-1\}}.$$

Given all above, for a specific user $u$, it is easy to write out the negative log-likelihood:

$$\mathcal{L}(\mathcal{O}^u, \mathcal{Y}^u | \boldsymbol{s}^u, \lambda^u) = - \sum_{(i,j) \in \mathcal{O}_u} \log P\{y_{ij}^u\}.$$
$$s.t. \ \boldsymbol{s}^u = \boldsymbol{c}_s + \boldsymbol{p}_s^u, \ \lambda^u = c_\lambda + p_\lambda^u, \ \lambda^u \ge \delta, \ c_\lambda \ge \delta. \tag{6}$$

In the constraints we use $\lambda^u \ge \delta$, $c_\lambda \ge \delta$, where $\delta > 0$, as closed and convex approximations of the positivity constraints $\lambda^u > 0, c_\lambda > 0$. The benefit to employ the relaxations are two-fold: 1) The closed domain constraints induce closed-form solution; 2) The threshold $\delta$ improves the quality of the solution to avoid ill-conditioned cases being too close to zero.

Obviously, the personalized bias could not grow arbitrarily large. More reasonably, only highly personalized users have a significant bias $\boldsymbol{p}_s^u$ and $p_\lambda^u$, while the majority of the mass tends to have smaller or even zero biases. If we denote $\boldsymbol{P}_s = \{\boldsymbol{p}_s^u : u = 1, \cdots, U\}$ and $\boldsymbol{P}_\lambda = \{p_\lambda^u : u = 1, \cdots, U\}$, this means that $(\boldsymbol{P}_s, \boldsymbol{P}_\lambda)$ satisfies group sparsity, then we add a group lasso penalty to the loss function $J_\mu(\boldsymbol{P}_s, \boldsymbol{P}_\lambda)$, which is in the form:

$$J_\mu(\boldsymbol{P}_s, \boldsymbol{P}_\lambda) = \mu \sum_u \left\| \begin{bmatrix} \boldsymbol{p}_s^u \\ p_\lambda^u \end{bmatrix} \right\|, \quad \mu > 0, \tag{7}$$

where $\mu$ is a regularization parameter. Such a structural penalty (7) can identify abnormal users $u$ whose $\boldsymbol{p}_s^u$ and $p_\lambda^u$ are nonzero. These non-zero terms increase the penalty function. However the corresponding reduction of loss function $\mathcal{L}(\mathcal{O}^u, \mathcal{Y}^u | \boldsymbol{s}^u, \lambda^u)$ must dominate the increasing penalty so as to minimize the overall objective function. In this sense, the abnormal users capture the strong signals for individualized biases. However, it ignores the possibility that weak signals could also induce individualized biases. Such signals help to decrease the loss, but the reduction of loss is not strong enough to cover the penalty term. This motivates us to propose a variable splitting scheme to simultaneously embrace strong and weak patterns. Specifically, we model the overall signal $(\boldsymbol{p}_s^u, p_\lambda^u)$ as the sum of the strong signals $(\boldsymbol{\Gamma}_s^u, \Gamma_\lambda^u)$ and weak signals $(\boldsymbol{\Delta}_s^u, \Delta_\lambda^u) = (\boldsymbol{p}_s^u, p_\lambda^u) - (\boldsymbol{\Gamma}_s^u, \Gamma_\lambda^u)$. The group lasso penalty is exhibited on the strong signals. Moreover, we give the weak signals an

$\ell_2$ penalty in the form: $\mathcal{S}_\nu(\boldsymbol{\Gamma}, \boldsymbol{P}) = \frac{1}{2\nu} \sum_u \left\| \begin{bmatrix} \boldsymbol{\Gamma}_s^u \\ \Gamma_\lambda^u \end{bmatrix} - \begin{bmatrix} \boldsymbol{p}_s^u \\ p_\lambda^u \end{bmatrix} \right\|^2$, to avoid it being arbitrarily large.

Denote the parameter set as $\Theta = \{\boldsymbol{P}_s, \boldsymbol{P}_\lambda, \boldsymbol{\Gamma}_s, \boldsymbol{\Gamma}_\lambda, \boldsymbol{c}_s, c_\lambda\}$. Define $\boldsymbol{\Gamma}_s = \{\boldsymbol{\Gamma}_s^u : u = 1, \cdots, U\}$ and $\boldsymbol{\Gamma}_\lambda = \{\Gamma_\lambda^u : u = 1, \cdots, U\}$, the loss function is defined as:

$$\min_{\Theta} \quad \sum_u \mathcal{L}(\mathcal{O}^u, \mathcal{Y}^u | \boldsymbol{s}^u, \lambda^u) + \mathcal{S}_\nu(\boldsymbol{\Gamma}, \boldsymbol{P}) + J_\mu(\boldsymbol{\Gamma}_s, \boldsymbol{\Gamma}_\lambda)$$

$$s.t. \quad \boldsymbol{s}^u = \boldsymbol{c}_s + \boldsymbol{\Gamma}_s^u, \; \lambda^u = c_\lambda + \Gamma_\lambda^u, \tag{8}$$

$$\lambda^u \geq \delta, \; c_\lambda \geq \delta.$$

Instead of directly solving the above-mentioned problem, we adopt the Split Linearized Bregman Iterations which we call individualized Split LBI (`iSplitLBI`), which gives rise to a regularization path where both the model parameters and hyper-parameters are simultaneously evolved. The $(k+1)$-th iteration on such a path is given as:

$$\begin{pmatrix} \boldsymbol{c}_s^{u,k+1} \\ c_\lambda^{u,+1} \end{pmatrix} = \mathbf{Prox}_{J_c}\left( \begin{pmatrix} \boldsymbol{c}_s^{u,k} \\ c_\lambda^{u,k} \end{pmatrix} - \kappa \alpha_k \nabla_{\mathbf{c}} \mathcal{L}(\Theta^k) \right), \forall u \in \mathcal{U} \tag{9a}$$

$$\begin{pmatrix} \boldsymbol{P}_s^{k+1} \\ \boldsymbol{P}_\lambda^{k+1} \end{pmatrix} = \mathbf{Prox}_{J_{\boldsymbol{P}}}\left( \begin{pmatrix} \boldsymbol{P}_s^k \\ \boldsymbol{P}_\lambda^k \end{pmatrix} - \kappa \alpha_k \nabla_{\boldsymbol{P}} \mathcal{L}(\Theta^k) \right), \tag{9b}$$

$$\begin{pmatrix} \boldsymbol{Z}_s^{k+1} \\ \boldsymbol{Z}_\lambda^{k+1} \end{pmatrix} = \begin{pmatrix} \boldsymbol{Z}_s^k \\ \boldsymbol{Z}_\lambda^k \end{pmatrix} - \alpha_k \nabla_{\boldsymbol{\Gamma}} \mathcal{L}(\Theta^k), \tag{9c}$$

$$\begin{pmatrix} \boldsymbol{\Gamma}_s^{k+1} \\ \boldsymbol{\Gamma}_\lambda^{k+1} \end{pmatrix} = \kappa \cdot \mathbf{Prox}_{J_\mu}\left( \begin{pmatrix} \boldsymbol{Z}_s^k \\ \boldsymbol{Z}_\lambda^k \end{pmatrix} \right), \; \mu = 1, \tag{9d}$$

where the initial choice $c_\lambda^{u,0} = 1 \in \mathbb{R}^1$, $\mathbf{c}_s^{u,0} = 0 \in \mathbb{R}^p$, $\boldsymbol{P}_s^0 = \boldsymbol{Z}_s^0 = 0 \in \mathbb{R}^{U \times p}$, $\boldsymbol{P}_\lambda^0 = \boldsymbol{Z}_\lambda^0 = 0 \in \mathbb{R}^U$, parameters $\kappa > 0$, $\alpha > 0$, $\nu > 0$, and the proximal map associated with a convex function $h$ is defined by $\mathbf{Prox}_h(\boldsymbol{z}) = \arg\min_{\boldsymbol{x}} \|\boldsymbol{z} - \boldsymbol{x}\|^2 / 2 + h(\boldsymbol{x})$. The $J_c(c_\lambda)$ and $J_{\boldsymbol{P}}(\boldsymbol{P}_\lambda)$ are denoted as the indicator function for the set $c_\lambda \geq \delta$ and $\lambda^u \geq \delta$ respectively (an indicator function of a set is 0 when the input variable is in the set, otherwise it is $+\infty$). Hence, at each step, the first two steps give a projected gradient descent of $c_\lambda$ and $\boldsymbol{P}_\lambda$, which makes the variables feasible.

The `iSplitLBI` algorithm generates a regularized solution path of dense estimators $(c_s^k, c_\lambda^k, \boldsymbol{P}_s^k, \boldsymbol{P}_\lambda^k)$ and sparse estimators $(\tilde{\boldsymbol{P}}_s^k, \tilde{\boldsymbol{P}}_\lambda^k)$. These sparse estimators could be obtained by projecting $(\boldsymbol{P}_s, \boldsymbol{P}_\lambda)$ onto the support set of $(\boldsymbol{\Gamma}_s, \boldsymbol{\Gamma}_\lambda)$, respectively. Along the path, the stopping time at $\tau_k = \sum_{i=1}^k \alpha_i$ in this algorithm plays the same role as the regularization parameter in the lasso problem. In fact, Eq.(9a)-(9d) describes one iteration of the optimization process, which is actually a discretization of a dynamical system shown in [10]. Such a dynamical system is known as inverse scale spaces [1, 21, 9], leveraging a regularization path consisting of sparse models at different levels from the null to the full. At iteration $k$, the cumulative time $\tau_k$ can be regarded as the inverse of the Lasso regularization parameter (here roughly $\tau_k \sim 1/\mu$): the larger is $\tau_k$, the smaller is the regularization and hence the more nonzero parameters enter the model. Following the dynamics, the model gradually grows from sparse to dense models with increasing complexity. In particular as $\tau_k \to \infty$, the dynamics may reach some over-fitting models when noise exists like our case, equivalent to a full model in generalized Lasso of minimal regularization. To prevent such over-fitting models in noisy applications, we adopt an early stopping strategy to find an optimal stopping time by cross validation.

Moreover, the $\nu$ also plays an important role in the model. When $\nu \to 0$, only sparse strong signals (features) are kept in models, then the `iSplitLBI` reduces to LBI algorithm, which is shown to reach model selection consistency under nearly the same condition as LASSO for linear models [21]. Recently, it is shown in [10] that the model selection consistency can also hold even under non-linear models. With a finite value of $\nu$, it is shown in [8, 9] that the sparse estimator enjoys improved model selection consistency. Moreover, equipped with the variable splitting scheme, the finite value of $\nu$ enables the overall signals (here $\boldsymbol{P}$) to capture features ignored by the strong (sparse) signals. It has been shown in the literature (e.g. [24, 32]), which coincides with our discussion, that such kinds of features can improve prediction in various tasks. Now we note the following implementation details for `iSplitLBI`. The hyper-parameter $\kappa$ is a damping factor which determines the bias of the sparse estimators, a bigger $\kappa$ leading to less biased estimators (bias-free as $\kappa \to \infty$). The hyper-parameter $\alpha_k$ is the step size which determines the precise of the path, with a large $\alpha_k$ rapidly traversing a coarse-grained path. However one has to keep $\alpha_k \kappa$ small to avoid possible oscillations of the paths, e.g. $\alpha_k \kappa \leq \frac{2}{\|\nabla^2 \mathcal{L}(\Theta^k)\|_2^2}$. The default choice in this paper is $\frac{1}{\kappa \|\nabla^2 \mathcal{L}(\Theta^k)\|_2^2}$ as a tradeoff between performance and computation cost.

## 2.3 Decomposition Property of iSplit LBI

By virtue of the variable splitting term, the dense parameter $\boldsymbol{P}$ enjoys a specific orthogonal decomposition property, as is shown in Fig.2:

$$\boldsymbol{P} = \boldsymbol{P}_{abn} \oplus \boldsymbol{P}_{per} \oplus \boldsymbol{P}_{ran}.$$

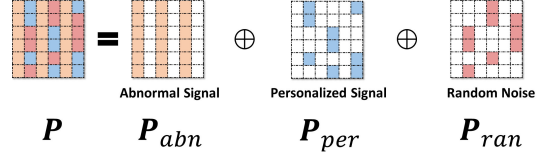

Figure 2: Decomposition of personalized parameters.

(1) $\boldsymbol{P}_{abn}$ is simply $\tilde{\boldsymbol{P}}$, i.e., the projection of $\boldsymbol{P}$ on the support set of $\boldsymbol{\Gamma}$. In other words, $P_{abn_{ij}} = P_{ij}$ if $\Gamma_{ij} \neq 0$, and $P_{abn_{ij}} = 0$ otherwise. Users corresponding to the non-zero columns of $\boldsymbol{P}_{abn}$ have significant biases toward the popular scores $\boldsymbol{c}_s$ and the common threshold $c_\lambda$. Thus the structure of $\boldsymbol{P}_{abn}$ could tell us who is an abnormal user in the crowd. In this sense, we refer to $\boldsymbol{P}_{abn}$ as the abnormal signal. This corresponds to the strong signals in the last subsection. (2) Among the remainder of such projection, $\boldsymbol{P}_{per}$ stands for the elements having a significant magnitude than random noise. This component drives the dense parameter $\boldsymbol{P}$ further away from the sparse parameter $\tilde{\boldsymbol{P}}$. According to the discussion in the previous subsection, this component takes into consideration the weak signals that help to further reduce the loss function. In this sense, including $\boldsymbol{P}_{per}$ brings better performance to $\boldsymbol{P}$. (3) The remaining entries in $\boldsymbol{P}$ are referred to as $\boldsymbol{P}_{ran}$, i.e., the random noises, which are inevitable due to the randomness of the data.

With all above, we present a compatible framework for both model prediction and model selection: (1) The strong signal $(\boldsymbol{P}_s, \boldsymbol{P}_\lambda)$ contains all the personalized biases which is a better choice for model prediction; (2) $(\boldsymbol{\Gamma}_s, \boldsymbol{\Gamma}_\lambda)$ and $\boldsymbol{P}_{abn}$ exclude the weak and dense personalized signals in the overall signals, which makes it a natural choice of abnormal user identification using model selection. This motivates us to take advantage of the support set of $\tilde{\boldsymbol{P}}$ to detect abnormal users, while utilizing $\boldsymbol{P}$ for prediction.

## 3 Experiments

### 3.1 Simulated Study

**Settings**. We validate our algorithm on simulated data with $n = |V| = 20$ items and $U = 50$ annotators. We first generate the true common ranking scores $\boldsymbol{c}_s \sim \mathcal{N}(0, 5^2)$. Then each annotator has a probability $p_1 = 0.2$ to have a nonzero $\boldsymbol{p}_s^u$. Those nonzero $\boldsymbol{p}_s^u$s are drawn randomly from $\mathcal{N}(0, 5^2)$. If $\boldsymbol{p}_s^u$ is nonzero, we generate $p_\lambda^u$ as $p_\lambda^u \sim c_\lambda * \mathcal{U}(-0.5, 0.5)$, otherwise we simply set $p_\lambda^u = 0$, where $c_\lambda = 1.5$. At last, we draw $N^u$ samples for each user randomly following the Bradley-Terry model. The sample number $N^u$ uniformly spans $[N_1, N_2] = [200, 400]$. Finally, we obtain a multi-edge graph with ties annotated by 50 annotators.

**Abnormal User Detection**. In this part, we validate abnormal user detection ability of `iSplitLBI` with visualization analysis. As we have stated, the support set of $\boldsymbol{P}$ (or $\tilde{\boldsymbol{P}}$ equivalently) implies the abnormal users. In this sense, we visualize the $\tilde{\boldsymbol{P}}$ (the ground-truth parameters) and $\tilde{\boldsymbol{P}}_0$ (the estimated parameters) in Fig.4 (a)-(b), whereas we visualize the magnitude of $\tilde{\boldsymbol{P}}$ (i.e. $|\tilde{\boldsymbol{P}}|$) and $\tilde{\boldsymbol{P}}_0$ (i.e. $|\tilde{\boldsymbol{P}}_0|$) in Fig.4 (c)-(d). Although the magnitude of $\tilde{\boldsymbol{P}}_0$ tends to be smaller than the true parameter, the results in Fig.4 (a)-(b) clearly suggest a perfect detection of the abnormal users.

Furthermore, Fig.5 shows the $L_2$-distance between each user's individualized ranking (i.e., $\boldsymbol{s}^u$) and the common ranking (i.e., $\boldsymbol{c}_s$), $\|\boldsymbol{s}^u - \boldsymbol{c}_s\|$. Clearly one can see the abnormal users we detected all exhibit larger L2-distance with the common ranking compared with other users. This indicates that these 13 abnormal users detected are those with large deviations from the population's opinion.

**Prediction Ability**. After showing the successful detection of abnormal users, in the following, we will exhibit the prediction ability of the proposed iSplit LBI method.

(1) Evaluation metrics: We measure the experimental results via two evaluation criteria, i.e., Macro-F1, and Micro-F1 over the three classes -1,0,1, which take both precision and recall into account. Note that the larger the value of Micro-F1 and Macro-F1, the better the performance. For more details, please refer to [31].

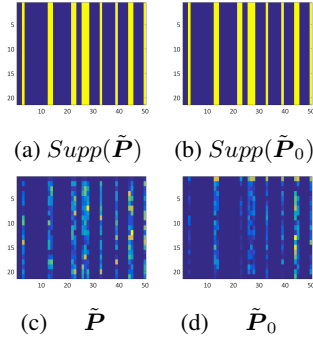

(a) $Supp(\tilde{\boldsymbol{P}})$ (b) $Supp(\tilde{\boldsymbol{P}}_0)$

(c) $\tilde{\boldsymbol{P}}$ (d) $\tilde{\boldsymbol{P}}_0$

Figure 4: Visualization of the parameters.

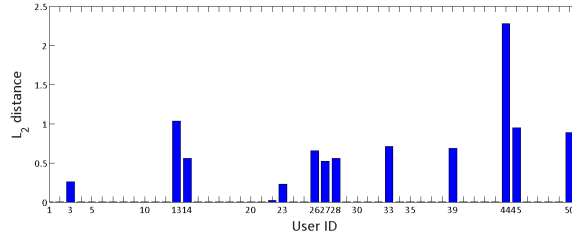

Figure 5: Detected abnormal users.

(2) Competitors: We employ two competitors that share most of the problem settings with `iSplitLBI`. i) the `α-cut algorithm` [3] is an early trial of common partial ranking. Since $\alpha$-cut is an ensemble-based algorithm, its performance depends on the choice of weak learners. Consequently, we compare our proposed algorithm with the $\alpha$-cut algorithm where different types of such weak learners and regularization schemes are adopted. Regarding the parameter-tuning of the weak learners in $\alpha$-cut, we tune the coefficients for Ridge/LASSO regularization from the range $\{2^{-15}, 2^{-13}, \cdots, 2^{-5}\}$ and the best parameters are picked out through a 5-fold cross-validation on the training set. ii) a most recently developed `margin-based MLE method` [29] where `Uniform, Bradley-Terry, and Thurstone-Mosteller` models are considered, respectively.

(3) Qualitative Results: Tab.1 shows the corresponding performance of our proposed algorithms and the competitors. In this table, the second column shows the weak learners and regularization terms employed in `α-cut` and three models proposed in `MLE-based` algorithm. Specifically, LR represents for logistics regression, SVM stands for the Support Vector Machine method, LS stands for the method of least squares while SVR stands for the Support Vector Regression method. For regularization, we employ the Ridge and LASSO regularization terms. Here we split the data into a training set ($80\%$ of each user's pairwise comparisons) and a testing set (the remaining $20\%$). To ensure the statistical stability, we repeat this procedure 20 times. It is easy to see that iSplit LBI significantly outperforms the other two competitors with an average of $0.834 \pm 0.005$ in Micro-F1 and $0.761 \pm 0.007$ in Macro-F1 due to its individualized property.

| types | algorithms | (a) Micro-F1 | | | | (b) Macro-F1 | | | |
|-------|-----------|-----|--------|-----|------|-----|--------|-----|------|
| | | min | median | max | std | min | median | max | std |
| $\alpha$-cut | LRLasso | .216 | .345 | .365 | .033 | .238 | .323 | .364 | .036 |
| | LRRidge | .319 | .347 | .380 | .018 | .210 | .338 | .392 | .061 |
| | SVMlasso | .318 | .340 | .367 | .012 | .216 | .305 | .401 | .051 |
| | SVMRidge | .294 | .349 | .367 | .016 | .198 | .334 | .429 | .077 |
| | LSLasso | .306 | .344 | .368 | .016 | .206 | .347 | .413 | .044 |
| | LSRidge | .320 | .346 | .368 | .013 | .222 | .325 | .410 | .051 |
| | SVRlasso | .329 | .347 | .377 | .013 | .221 | .357 | .448 | .057 |
| | SVRRidge | .312 | .336 | .378 | .019 | .220 | .346 | .436 | .050 |
| MLE | Un | .599 | .622 | .660 | .014 | .588 | .605 | .631 | .011 |
| | BT | .772 | .801 | .839 | .015 | .628 | .660 | .679 | .015 |
| | TM | .767 | .799 | .820 | .016 | .608 | .637 | .669 | .016 |
| Ours | Ours | .825 | **.834** | .841 | .005 | .747 | **.761** | .774 | .007 |

Table 1: Experimental results on simulated dataset.

## 3.2 Human Age

**Dataset**. In this dataset, 25 images from human age dataset FG-NET [1] are annotated by a group of volunteers on ChinaCrowds platform. The annotator is presented with two images and given a choice of which one is older (or difficult to judge). Totally, we obtain 9589 feedbacks from 91 annotators.

**Qualitative Results**. Tab.2 shows the corresponding performance of our proposed algorithms and the competitors. We can easily find that our proposed algorithm significantly outperforms the other two competitors in terms of both Micro-F1 and Macro-F1. Moreover, Fig.6 (a) shows the $L_2$-distance between selected users' (i.e., the top 10% and bottom 10% in the regularization path) individualized ranking and the common ranking. Clearly one can see that users jumped out earlier (i.e.,

[1]http://www.fgnet.rsunit.com/

the top 10% marked with pink) show larger $L_2$-distance, thus are those with large deviation from the population's opinion and can be treated as abnormal users. On the contrary, users jumped out later (i.e., the bottom 10% marked with blue) tend to have smaller or even zero $L_2$-distance.

## 3.3 WorldCollege Ranking

**Dataset**. We now apply the proposed method to the world college ranking dataset, which is composed of 261 colleges. Using the Allourideas crowdsourcing platform, a total of 340 random annotators with different backgrounds from various countries (e.g., USA, Canada, Spain, France, Japan, China, etc.) are shown randomly with pairs of these colleges and asked to decide which of the two universities is more attractive to attend. If the voter thinks the two colleges are incomparable, he/she can choose the third option by clicking "I cannot decide". Finally, we obtain a total of 11012 feedbacks, among which 9409 samples are pairwise comparisons with clear opinions (i.e., 1/-1) and the remaining 1603 are samples records with voter clicking "I cannot decide" (i.e., 0).

|  | | (a) Micro-F1 | | | | (b) Macro-F1 | | | |
|---|---|---|---|---|---|---|---|---|---|
| types | algorithms | min | median | max | std | min | median | max | std |
| $\alpha$-cut | LRLasso | .428 | .443 | .458 | .008 | .327 | .358 | .381 | .016 |
|  | LRRidge | .422 | .443 | .457 | .008 | .330 | .364 | .387 | .015 |
|  | SVMlasso | .422 | .442 | .463 | .010 | .331 | .355 | .371 | .013 |
|  | SVMRidge | .424 | .443 | .457 | .008 | .330 | .351 | .379 | .014 |
|  | LSLasso | .423 | .441 | .455 | .008 | .335 | .361 | .383 | .014 |
|  | LSRidge | .426 | .442 | .464 | .009 | .335 | .365 | .398 | .014 |
|  | SVRlasso | .418 | .431 | .449 | .008 | .333 | .364 | .397 | .014 |
|  | SVRRidge | .422 | .432 | .450 | .007 | .337 | .367 | .381 | .012 |
| MLE | Uni. | .692 | .705 | .738 | .012 | .589 | .606 | .641 | .012 |
|  | BT | .731 | .741 | .755 | .008 | .599 | .628 | .647 | .012 |
|  | TM | .728 | .739 | .756 | .008 | .603 | .623 | .647 | .012 |
| Ours | Ours | .765 | **.779** | .791 | .007 | .680 | **.694** | .712 | .010 |

Table 2: Experimental results on Human Age dataset.

**Qualitative Results**. Tab.3 shows the comparable results on the college dataset. It is easy to see that our proposed algorithm again achieves better Micro-F1 and Macro-F1 with a large margin than all the $\alpha$-cut and MLE-based variants. To investigate the reason behind this, we further compare our proposed algorithm with the MLE-based algorithms in terms of fine-grained precision, recall performances on label $\{-1, 0, 1\}$ in Fig.6 (c). For labels -1 and 1, the performance improvement is relatively small, whereas a sharp improvement is highlighted for label 0. This suggests that the major contribution of the overall improvements of our proposed algorithm comes from its strength to recognize the incomparable pairs, which is exactly the main

|  | | (a) Micro-F1 | | | | (b) Macro-F1 | | | |
|---|---|---|---|---|---|---|---|---|---|
| types | algorithms | min | median | max | std | min | median | max | std |
| $\alpha$-cut | LRLasso | .318 | .350 | .408 | .026 | .328 | .349 | .367 | .011 |
|  | LRRidge | .325 | .352 | .408 | .023 | .323 | .348 | .364 | .010 |
|  | SVMlasso | .325 | .343 | .404 | .029 | .333 | .345 | .371 | .009 |
|  | SVMRidge | .327 | .354 | .402 | .025 | .327 | .345 | .365 | .010 |
|  | LSLasso | .305 | .320 | .377 | .016 | .331 | .342 | .362 | .010 |
|  | LSRidge | .331 | .345 | .403 | .023 | .334 | .345 | .365 | .009 |
|  | SVRlasso | .306 | .328 | .378 | .018 | .327 | .346 | .363 | .010 |
|  | SVRRidge | .323 | .348 | .402 | .023 | .323 | .350 | .361 | .012 |
| MLE | Uni. | .521 | .536 | .550 | .009 | .482 | .496 | .514 | .009 |
|  | BT | .539 | .552 | .565 | .008 | .496 | .513 | .526 | .010 |
|  | TM | .539 | .551 | .565 | .008 | .496 | .511 | .526 | .010 |
| Ours | Ours | .637 | **.649** | .663 | .008 | .608 | **.645** | .674 | .016 |

Table 3: Experimental results on College dataset.

pursuit of this paper. Moreover, similar to the human age dataset, we also plot the $L_2$ distance between the top/bottom 10% users' individualized ranking and the common ranking and similar phenomenon occurs on this dataset, as is shown in Fig.6 (b). Again, we see a significant difference between the recognized most individualized rankers and the least individualized rankers.

## 4 Conclusions

In this paper, we propose a novel method called iSplitLBI which is capable of simultaneously predicting personalized rankings with ties and detecting the abnormal users in the crowd. To tackle the personalized deviations of the scores, a hierarchical decomposition of the model parameters is designed where both the popular opinions and the individualized effects are taken into consideration. In what follows, a specific variable splitting scheme is adopted to separate the functionality of model prediction and abnormal user detection. Experiments on both simulated examples and real-world applications together demonstrate the effectiveness of the proposed method.

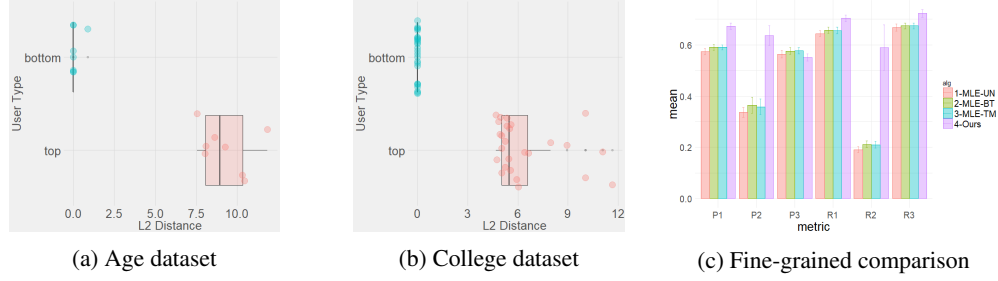

(a) Age dataset        (b) College dataset        (c) Fine-grained comparison

Figure 6: (a)-(b) The $L_2$ distance between individualized ranking scores and common ranking scores of selected users on age and college datasets. (c) Fine-grained comparison on college dataset. P1, P2, P3 represent the precision for class -1, 0, 1, respectively; while R1, R2, R3 represent the corresponding recalls, respectively.

## Acknowledgments

This work was supported in part by the National Key R&D Program of China (Grant No. 2016YFB0800403), in part by National Natural Science Foundation of China: 61620106009, U1636214, 61836002, U1803264, U1736219, 61672514 and 61976202, in part by National Basic Research Program of China (973 Program): 2015CB351800, in part by Key Research Program of Frontier Sciences, CAS: QYZDJ-SSW-SYS013, in part by the Strategic Priority Research Program of Chinese Academy of Sciences, Grant No. XDB28000000, in part by Peng Cheng Laboratory Project of Guangdong Province PCL2018KP004, in part by Beijing Natural Science Foundation (4182079), in part by Youth Innovation Promotion Association CAS, and in part by Hong Kong Research Grant Council (HKRGC) grant 16303817.

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
