[Reviews · NeurIPS 2019]

Reviewer 1



1) This paper is clearly written. The motivating example in its introduction makes me believe that tie-aware ranking is crucial for crowdsourcing problems. 2) Traditional methods merely adopt the strong signal(sparse parameter) for model prediction and structural learning. Different from this routine, their proposed method explicitly separates the strong signals and weak signals, then uses strong signals to learn a semantic structure as the outlier indicator and combines both the weak and strong signals to do a fine-grained prediction. As pointed out in the work,its helps to decouple the model selection and model prediction process.

Reviewer 2



This paper proposes a novel and unified model for individualized learning, partial ranking, and abnormal detection, which simultaneously solves important problems in personalized learning. In general, it is well-written with a good organization and easy to follow. I think this work could benefit our community in its style as it blends intuitions with solid mathematical machinery. This is what I hope to see most in the application papers. As I have mentioned in the contribution part, I particularly enjoy the optimization rules induced by the split-LBI method, which provides a dynamic way to learn and select the correct model.

Reviewer 3



The formulation of annotators’ ranking decision, i.e. integrating personalized preference, abnormal deviations, common score, and random noise simultaneously from a probabilistic view, is well-designed and novel. The proposed model can give user-specific partial ranking prediction and abnormal user detection simultaneously, which is an early trail in the direction. It is also pleased to see that iSplitLBI achieves reasonable performance in discovering incomparable pairs. The proposed model unites simplicity and effectiveness, which makes it possible to be widely adopted for future work.

[Author Response · NeurIPS 2019]

# Response Letter of "iSplit LBI: Individualized Partial Ranking with Ties via Split LBI" ID 2161

We thank all the reviewers for your time in reviewing this paper and also for your suggestive comments. We would like to make the following clarifications.

## To Reviewer 1

**Q1**: **i) More explanation for Eq.(9). ii) why $\alpha^k$ is a regularization parameter? iii) why early-stopping is necessary?** i) Eq.(9a)-(9d) describes one iteration of the optimization process, which is actually a discretization of a dynamic system Ref.[20] in paper or [3] below. ii) Such dynamics are known as inverse scale spaces [1, 2], leveraging a regularization path consisting of sparse models at different levels from the null to the full. At iteration $k$, the cumulating time $\tau_k = \sum_{i=1}^{k} \alpha_i$ can be regarded as the inverse of the generalized Lasso regularization parameter $\lambda$: the larger is $\tau_k$, the smaller is the regularization and hence the more nonzero parameters enter the model. iii) Following the dynamic system, the model gradually grows from sparse to dense models with increasing complexity. In particular as $\tau_k \to \infty$, the dynamics may reach some over-fitting models when noise exists like our case, equivalent to a full model in generalized Lasso of minimal regularization. To prevent such overfitting models in noisy applications, early stopping is necessary to find an optimal model by cross validation. We will add detailed remarks if accepted.

**Q2**: **Macro- and Micro- Fs.** Yes, they are the same as the metrics used in the multi-label setting. Please see our Ref.[28] in the main paper for more details.

**Q3**: **Some typos to be corrected.** We will carefully fix all the typos and improve the grammar in the new version if accepted.

## To Reviewer 2

**Q1**: **More explanations on the inequality constraints.** As pointed out by the reviewer, we use a closed domain approximation of the original open domain constraints to improve the stability in the following two aspects. i) The convergence property of the gradient projection method relies on the projection theorem. Unfortunately, this theorem holds only for closed and convex feasible sets. The approximation makes the original convex set closed, thus obeys the projection theorem. ii) Such a closed domain constraint induces a closed-form solution and improves the quality of the solution to avoid ill-conditioned cases being too close to zero.

**Q2**: **The setting for human age and WorldCollege ranking.** They follow the same settings with the simulated study. We will clarify it if accepted.

**Q3**: **Null users in Fig.5.** The users that do not appear in Fig.5 have almost zero distance toward the common parameter. This validates the effectiveness of our proposed method to select abnormal users from the crowd. We will improve the descriptions if accepted.

## To Reviewer 3

**Q1**: **Add some discussions on the decomposition property.** We call $P_{abn}$ the strong signals, since it corresponds to the non-zero users in the sparse parameter $\Gamma$. Such strong signals induce a sharp reduction of the loss and dominate the sparse penalty function. The residual between $P$ and $P_{abn}$ consists of the parameters that fail to give a significant reduction of the loss function, that can be of either weak signals (the users are not of much difference to the common) or random noise, as shown in Sec. 2.3.

**Q2**: **Missing users in Fig.5.** Please see the answers for Q3 of Reviewer 2.

## References

[1] M. Burger, M. Möller, M. Benning, and S. Osher. An adaptive inverse scale space method for compressed sensing. *Mathematics of Computation*, 82(281):269–299, 2013.

[2] M. Burger, S. Osher, J. Xu, and G. Gilboa. Nonlinear inverse scale space methods for image restoration. In *International Workshop on Variational, Geometric, and Level Set Methods in Computer Vision*, pages 25–36. Springer, 2005.

[3] C. Huang and Y. Yao. A unified dynamic approach to sparse model selection. In *International Conference on Artificial Intelligence and Statistics*, pages 2047–2055, 2018.


[Meta-Review · NeurIPS 2019]

This is a worthy paper that presents a significant advance in an area that has seen relatively little recent work.